# Targeted Blood Plasma Proteomics and Hemostasis Assessment of Post COVID-19 Patients with Acute Myocardial Infarction

**DOI:** 10.3390/ijms24076523

**Published:** 2023-03-30

**Authors:** Anna Kalinskaya, Daria Vorobyeva, George Rusakovich, Elena Maryukhnich, Alexandra Anisimova, Oleg Dukhin, Antonina Elizarova, Oxana Ivanova, Anna Bugrova, Alexander Brzhozovskiy, Alexey Kononikhin, Evgeny Nikolaev, Elena Vasilieva

**Affiliations:** 1Laboratory of Atherothrombosis, Cardiology Department, A.I. Evdokimov Moscow State University of Medicine and Dentistry, 127473 Moscow, Russia; 2I.V. Davydovsky Moscow City Clinical Hospital, Moscow Department of Healthcare, 117463 Moscow, Russia; 3Emanuel Institute for Biochemical Physics, Russian Academy of Science, 119991 Moscow, Russia; 4Center for Molecular and Cellular Biology, Skolkovo Institute of Science and Technology, 121205 Moscow, Russia

**Keywords:** post-COVID, SARS-CoV-2, COVID-19, acute myocardial infarction, proteomics analysis, hemostasis

## Abstract

The molecular mechanisms underlying cardiovascular complications after the SARS-CoV-2 infection remain unknown. The goal of our study was to analyze the features of blood coagulation, platelet aggregation, and plasma proteomics in COVID-19 convalescents with AMI. The study included 66 AMI patients and 58 healthy volunteers. The groups were divided according to the anti-N IgG levels (AMI post-COVID (*n* = 44), AMI control (*n* = 22), control post-COVID (*n* = 31), and control (*n* = 27)). All participants underwent rotational thromboelastometry, thrombodynamics, impedance aggregometry, and blood plasma proteomics analysis. Both AMI groups of patients demonstrated higher values of clot growth rates, thrombus size and density, as well as the elevated levels of components of the complement system, proteins modifying the state of endothelium, acute-phase and procoagulant proteins. In comparison with AMI control, AMI post-COVID patients demonstrated decreased levels of proteins connected to inflammation and hemostasis (lipopolysaccharide-binding protein, C4b-binding protein alpha-chain, plasma protease C1 inhibitor, fibrinogen beta-chain, vitamin K-dependent protein S), and altered correlations between inflammation and fibrinolysis. A new finding is that AMI post-COVID patients opposite the AMI control group, are characterized by a less noticeable growth of acute-phase proteins and hemostatic markers that could be explained by prolonged immune system alteration after COVID-19.

## 1. Introduction

Hypercoagulation and a high incidence of thrombotic complications in the acute phase of SARS-CoV-2 infection have been reported [1,2]. Another issue that has come to light during the pandemic is that of long-term cardiovascular and neurological consequences [3]. In particular, among patients convalescing from COVID-19, a greater risk of developing myocardial infarction (MI) has been demonstrated [4,5].

Although several possible mechanisms for the development of MI in COVID-19 over the long term (endothelial dysfunction, activation of the blood coagulation system, dysfunction of the immune system) have been proposed, at present the exact mechanisms for the development of this phenomenon remain unclear [6,7].

In consideration of the findings and problems mentioned above, the search for specific mechanisms of cardiovascular complications in the long-term period of COVID-19 is of great importance. Targeted proteomics is a promising technique to identify key links in the pathogenesis of post-COVID-19 [8,9]. The use of targeted proteomics with a validated assay of peptide standards and robust instrumentation, such as triple quadrupole, provides the results which have more potential for further translation in clinical practice. Moreover, the targeted approach allows the production of consistent results across different instrumental platforms, and thus allows biomarker validation using the exact same methods and workflows in independent cohorts and laboratories [10,11].

In the current study, we considered the protein and corresponding peptide panel from the BAK270 MRM assay [12]. The assay is developed for the analysis of potential protein biomarkers for cardiovascular disease, including 61 FDA-approved biomarkers in blood plasma. Moreover, the robustness of the current MRM assay for selected blood proteins was recently demonstrated [13]. We investigated the peculiarities of plasma coagulation, endogenous fibrinolysis, platelet function, and blood plasma proteomics, including levels of acute-phase proteins, components of the complement system, pro- and anticoagulant proteins, and proteins modifying the endothelium state in patients with acute myocardial infarction (AMI), recovered from SARS-CoV-2 infection.

## 2. Results

### 2.1. Clinical Data

The average demographic and clinical characteristics of the groups included in the study are presented in Table 1. Between patients with AMI, there are no significant differences in clinical data values (Table 1).

Compared with both control groups, AMI patients were significantly older and characterized by a more frequent history of arterial hypertension; in addition to this, compared with the control post-COVID group, the AMI post-COVID group was characterized by a higher prevalence of the male sex and a more frequent prevalence of smoking; compared with the control group, the AMI control group was characterized by higher values of BMI; compared with the control group, the AMI post-COVID group was characterized by a higher prevalence of the male sex, higher values of BMI, a higher prevalence of smoking (Table 1).

### 2.2. Laboratory Parameters

Compared with the control group, the AMI control group was characterized by higher values of hemoglobin, leukocytes, triglycerides, creatinine, INR, ALT, and AST, and by lower values of high-density lipoproteins (HDL), PT, APTT and AT-III (Table 2).

Compared with the control post-COVID group, the AMI post-COVID group was characterized by higher values of leukocytes, triglycerides, creatinine, INR, ALT, and AST, and by lower values of HDL, APTT, PT, and AT-III (Table 2).

Compared with the control group, the AMI post-COVID group was characterized by higher values of hemoglobin, leukocytes, triglycerides, creatinine, INR, ALT, AST, and by lower values of HDL, APTT, PT, and AT-III (Table 2).

### 2.3. Rotational Thromboelastometry

Compared with the AMI control group, the AMI post-COVID group was characterized by a lower value of mean clot firmness (MCF, mm, 59 [56; 64] vs. 56 [50; 60], *p* adj. = 0.078).

Compared with the control group, the AMI control group was characterized by a larger thrombus size (A15, mm, 49 [47; 53] vs. 54 [49; 60], *p* adj. = 0.064; A20, mm, 52 [49; 56] vs. 57 [53; 61], *p* adj. = 0.044; A30, mm, 55 [51; 57] vs. 58 [55; 63], *p* adj. = 0.047) and maximum clot firmness (MCF, mm, 56 [52; 58] vs. 59 [56; 64], *p* adj. = 0.044).

Comparisons of other parameters are presented in Appendix A.

### 2.4. Thrombodynamics

Compared with the AMI control group, the AMI post-COVID group was characterized by a lower value of clot density (D, arb units, 27,381.5 [25,505.75; 31,961.5] vs. 25,461.5 [22,630.75; 27,474.5], *p* adj. = 0.079).

Compared with the control group, the AMI control group was characterized by a higher value of clot density (D, arb units, 22,996.5 [21,665.75; 24,478.25] vs. 27,381.5 [25,505.75; 31,961.5], *p* adj. < 0.01), and a longer lag time (Tlag, min, 0.95 [0.8; 1] vs. 1.1 [1; 1.12], *p* adj. = 0.034).

Compared with the control post-COVID group, the AMI post-COVID group was characterized by higher values of clot growth rate (V, µm/min, 29.2 [26; 32.9] vs. 34.75 [30.96; 38.68], *p* adj. < 0.01), initial clot growth rate (Vi, µm/min, 52.6 [48.95; 56.45] vs. 63.4 [58.5; 66.38], *p* adj. < 0.01), stationary clot growth rate (Vst, µm/min, 29 [26; 32.71] vs. 34.7 [30.65; 38.6], *p* adj. < 0.01), and clot size (CS, µm, 1159 [1044.25; 1246.5] vs. 1384.5 [1245.12; 1450], *p* adj. < 0.01), and by shorter estimated lysis time (LTE, min, 28.2 [24.6; 36.5] vs. 21.65 [16.68; 31.45], *p* adj. = 0.073).

Compared with the control group, the AMI post-COVID group was characterized by higher values of initial clot growth rate (Vi, µm/min, 56.1 [53.08; 57.77] vs. 63.4 [58.5; 66.38], *p* adj. < 0.01), and clot density (D, arb units, 22,996.5 [21,665.75; 24,478.25] vs. 25,461.5 [22,630.75; 27,474.5], *p* adj. = 0.054).

Comparisons of other parameters are presented in Appendix A.

### 2.5. Impedance Aggregometry

There were no significant differences in impedance aggregometry values among patients with AMI.

Compared with the control group, the AMI control group was characterized by a lower value of arachidonic-acid-induced platelet aggregation (ASPI, AU × min, 60 [45; 65] vs. 40 [21; 47], *p* adj. = 0.054).

Compared with the control post-COVID group, the AMI post-COVID group was characterized by a lower value of arachidonic-acid-induced platelet aggregation (ASPI, AU × min, 58 [47.5; 68] vs. 26 [18; 40.75], *p* adj. < 0.01).

Compared with the control group, the AMI post-COVID group was characterized by a lower value of arachidonic-acid-induced platelet aggregation (ASPI, AU × min, 60 [45; 65] vs. 26 [18; 40.75], *p* adj. < 0.01).

Comparisons of other parameters are presented in Appendix A.

### 2.6. Proteomics

Targeted proteomics analysis for patients in all studied groups revealed 81 proteins that were reliably measured in all samples, and were connected to one of the processes described in the Appendix A: hemostasis, extracellular matrix (a structural component or modifier), modification of the state of endothelium, inflammation (except complement system as it was included in a separate group), complement system, lipid metabolism, calcification, and steroid hormone transport (a list of proteins in each group is presented in Appendix A). These processes were chosen because of their direct involvement in cardiovascular disease. 

Both the AMI control group (compared with the control group) and the AMI post-COVID group (compared with the control and control post-COVID groups) were characterized by elevated levels of acute-phase proteins (lipopolysaccharide-binding protein, C-reactive protein, haptoglobin), components of the complement system (complement C5, complement C9, complement factor B, and inhibitory complement factor I), procoagulant proteins (fibrinogen beta and gamma chains, coagulation factor IX), one protein with pro- and anticoagulant activities (beta-2-glycoprotein 1), and proteins modifying the state of endothelium (pigment epithelium-derived factor (PEDF), angiogenin).

Proteins elevated only in the AMI post-COVID group compared with the control groups included proteins involved in inflammation (attractin, phosphatidylinositol-glycan-specific phospholipase D), proteins of the complement system (complement C1q subcomponent subunit A, complement C4, complement C6, mannan-binding lectin serine protease 2A), and procoagulant proteins (coagulation factors XII and XIII B chain, fibronectin). By contrast, C4b-binding protein alpha-chain and L-selectin were decreased only in the AMI post-COVID group.

Proteins elevated only in the AMI control group compared with the control group included acute-phase protein leucine-rich alpha-2-glycoprotein, components of the complement system (complement C3, plasma protease C1 inhibitor), the regulator of hemostasis and the complement system (vitronectin), and anticoagulant vitamin-K-dependent protein S.

Compared with the AMI control group, the AMI post-COVID group was characterized by decreased levels of acute-phase lipopolysaccharide-binding protein, inhibitors of the complement system (C4b-binding protein alpha-chain, plasma protease C1 inhibitor), procoagulant fibrinogen beta-chain, anticoagulant vitamin-K-dependent protein S, and PEDF.

The main results of proteomics data comparisons between study groups are presented in Figure 1. The boxplots include proteins different between AMI control and AMI post-COVID groups: C4b-binding protein alpha-chain, fibrinogen beta-chain, lipopolysaccharide-binding protein, PEDF, plasma protease C1 inhibitor, vitamin K-dependent protein S. Based on their function, we decided to focus on proteins connected to hemostasis, acute-phase proteins, and components of the complement system. In the boxplots, we included proteins with these functions if they were different in at least one of the comparisons: AMI control vs. control, AMI post-COVID vs. control, AMI post-COVID vs. control post-COVID. One of the main functions of PEDF is negative regulation of angiogenesis, therefore we also added a protein with angiogenic function, which was also different in the comparisons mentioned above (angiogenin). Full comparisons of other proteins are presented in Appendix A. We tested the influence of gender, age, and BMI on the results obtained. It turned out that age and gender had no noticeable influence on the results (Spearman’s correlation coefficient > 0.5 or <−0.5, *p* adj. < 0.05). BMI correlated with the complement C3, complement factor B and complement factor I levels (Spearman’s correlation coefficient > 0.5, *p* adj. < 0.05) (See Appendix A).

Proteomics results for C-reactive protein and fibrinogen beta-chain were validated by immunoturbidimetry method and photo-optical detection method, respectively. Detailed protocol and results of validation are presented in the Appendix A.

Parameters describing clinical data (age, BMI) were analyzed separately (Appendix A). We also excluded some laboratory data (hemoglobin, platelets, total cholesterol, HDL, LDL, triglycerides, creatinine, AST, ALT), which were previously used for group comparison. We analyzed correlations between protein levels and parameters of hemostasis. We found correlations of acute-phase proteins (C-reactive protein, haptoglobin, hemopexin, leucine-rich alpha-2-glycoprotein, lipopolysaccharide-binding protein) and components of the complement system with parameters of blood clotting and/or procoagulant proteins in each group. Anticoagulant vitamin-K-dependent protein S correlated with the acute-phase proteins and/or components of the complement system in each group, and with the parameters of blood clotting in the AMI control and AMI post-COVID groups. The main correlations between the studied parameters are presented in Appendix A. Correlation matrices demonstrated only those parameters of hemostasis and proteins connected to hemostasis which correlated with proteins connected to inflammation (our division of proteins into functional groups is presented in Appendix A) and components of the complement system in at least one study group. Additionally, we included proteins connected to inflammation and components of the complement system if they correlated with parameters of hemostasis and proteins connected to hemostasis in at least one study group. The full correlation tables are presented in Appendix A, whereby each table corresponds to one study group. Full correlation tables include all 81 proteins and all the parameters of hemostasis analyzed in the study, and a *p* adjustment was done for this list of correlations. The control group showed correlations of parameters of fibrinolysis with complement C6, complement C8 alpha-chain, and C4b-binding protein alpha-chain. In the AMI control group, parameters of fibrinolysis correlated with haptoglobin. Any of these correlations were absent in the control post-COVID and AMI post-COVID groups (Appendix A). To illustrate this, we visualized differences in correlations between control and control post-COVID groups, and between AMI control and AMI post-COVID groups, which included only parameters of fibrinolysis, inflammatory proteins, and components of the complement system (Figure 2).

We performed clusterization of parameters of thrombodynamics and rotational thromboelastometry, results of coagulation blood tests, and proteins involved in inflammation and hemostasis, of components of the complement system, and of two proteins affecting endothelium (angiogenin, PEDF) (Figure 3 and Appendix A).

In the control group, we found cluster 1 predominantly consisting of components of the complement system and acute-phase proteins. Closely located clusters 4 and 6 in the control post-COVID group, and clusters 1 and 5 in the AMI control group, contained a very similar list of proteins with each other and cluster 1 in the control group. These clusters correspond to clusters 1 and 2 in the AMI post-COVID group, which are more remote from each other. Proteins similar in the listed clusters of the studied groups include acute-phase proteins (C-reactive protein, lipopolysaccharide-binding protein), proteins of the complement system (complement C3, complement factor B, complement factor I), and procoagulant proteins (fibrinogen beta-chain, fibrinogen gamma chain, vitronectin). Compared to clusters 1 and 5 in the AMI control group, clusters 1 and 2 in the AMI post-COVID group lack several components of the complement system, including complement C1q subcomponent subunit A, mannan-binding lectin serine protease 2A, complement C5, complement C8 alpha-chain, and plasma protease C1 inhibitor. Remarkably, in the AMI post-COVID group, components of the complement system formed another cluster (3) with lectin, and classical pathway proteins and components of the terminal stage of the complement system (complement C1q subcomponent subunit A, complement C1q subcomponent subunit B, mannan-binding lectin serine protease 2A, ficolin-2, complement C5, complement C6).

Proteins of which the elevation was similar in the AMI control group (compared to the control group) and the AMI post-COVID group (compared to the control and control post-COVID groups) were included in three separate clusters in the AMI control group, and 85% of them were included in clusters 1 and 5. In the AMI post-COVID group, these proteins were included in four separate clusters, and were predominantly distributed between clusters 1 and 2, and newly formed cluster 3. Proteins elevated only in the AMI control group, compared to the control group, were all included in clusters 1 and 5. Proteins elevated only in the AMI post-COVID group were almost evenly distributed between clusters 3, 4, and 5.

## 3. Discussion

The main purpose of our study presented here was to identify the peculiarities of the plasma protein profile and hemostasis in patients with AMI after SARS-CoV-2 infection. At first, we performed a protein analysis of AMI patients compared with the control groups, in order to identify the peculiarities of AMI itself. The alterations in AMI patients included elevated plasma levels of proteins connected to inflammation (acute-phase proteins, proteins of the complement system), prothrombotic proteins, one pro- and anticoagulant protein, and proteins affecting the endothelial layer (PEDF, angiogenin). Complement factor B and complement factor I positively correlated with the BMI level that differed between the study groups. Partially, the observed difference in complement factor B and complement factor I levels between AMI and control groups may be explained by this fact. On the other hand, the high BMI level is a well-known risk factor of the AMI development and can often accompany the AMI.

Lipopolysaccharide-binding protein and PEDF have previously been shown to be elevated in COVID-19-associated cardiovascular complications [14]. Elevated procoagulant protein levels in AMI patients are reported in the literature [15] and are in accordance with increased thrombus size, density, and growth rate parameters, as measured by thrombodynamics and rotational thromboelastometry. Elevated fibrinogen and coagulation factor IX levels have been shown to be a risk factor for AMI [16,17]. Acute-phase reaction and activation of the complement system in AMI patients were also described previously [18].

Moreover, we observed correlations of acute-phase proteins with parameters of hemostasis and prothrombotic proteins, which were present both in AMI groups and in the control post-COVID group, indicating the close relationship of inflammation and hemostasis in these groups of patients. Moreover, we saw correlations of the complement system proteins with the parameters of blood clotting and procoagulant proteins in all groups of patients. Interconnections between thrombosis, inflammation, and in particular complement activation, are described in the literature [19,20]. Both the complement and the hemostatic systems require proteolytic cleavage reactions for activation, providing a wide range of potential interactions between them. There is evidence that complement system components play a role in different stages of hemostasis, both in platelet activation and coagulation [21,22]. Moreover, the coagulation system has been shown to activate the complement system [19]. Finally, the complement and coagulation systems both have several common regulatory proteins [19].

We also revealed elevated levels of the proangiogenic protein angiogenin and the antiangiogenic PEDF in AMI groups compared with the control groups. Angiogenin elevation during AMI may be a response to hypoxia [23,24,25]. Moreover, angiogenin has been shown to be expressed as an acute-phase protein [26]. PEDF has also been shown to affect vascular permeability in different directions. On the one hand, it was found to maintain endothelial tight junctions after AMI [27,28]. On the other hand, PEDF was shown to increase vascular permeability in human umbilical vein endothelial cells (HUVEC) [29]. In addition, PEDF has cardioprotective effects during AMI [30]. Other actions of PEDF include anti-inflammatory and antithrombogenic activity, which may be a response to thrombosis and have beneficial effects [31,32].

The next step was to compare AMI groups, with and without previous SARS-CoV-2 infection. These groups demonstrated a difference in levels of inflammatory mediators. The AMI post-COVID group had a lower level of lipopolysaccharide-binding protein, and of the complement system inhibitors, as well as plasma protease C1 inhibitor and C4b-binding protein alpha-chain. Decreased levels of inflammatory mediators in the AMI post-COVID group compared to the AMI control group may reflect the chronic inflammation process in patients after SARS-CoV-2 infection [33], described previously. Because of this constant chronic inflammation in the AMI post-COVID group, in case of an acute cardiac event, such as MI, the immune system is not able to produce enough response, which is observed in AMI control patients.

Alterations in the inflammatory process in AMI post-COVID patients were also shown from clusterization of proteomics data and parameters of hemostasis. In each group, we found similar clusters formed by a list of proteins connected to inflammation and thrombosis. However, in the AMI post-COVID group, these clusters lacked several components of the complement system, and a unique cluster with the complement system proteins was formed.

We found lower plasma levels of fibrinogen beta-chains in the AMI post-COVID group in comparison with the AMI control group, corresponding to the lower optical density and firmness of the clot according to the hemostatic tests. Moreover, the levels of both antithrombotic vitamin-K-dependent protein S and the inhibitor of its activity, C4b-binding protein alpha-chains, were also decreased in the AMI post-COVID group, perhaps as a consequence of impaired clotting activity as a negative feedback loop, or a reduced inflammatory response. This is also confirmed by the correlations of vitamin-K-dependent protein S with the proteins involved in clotting and inflammation, especially in the complement system. The decreased level of PEDF in the AMI post-COVID group, compared with the AMI control group, may have different interconnections with the thrombotic event. First, a disrupted axis involving PEDF may be one of the causative factors of thrombosis due to antithrombogenic and anti-inflammatory actions of PEDF, or due to modulation of the integrity of the endothelial barrier. Second, it can be a consequence of impaired inflammatory response and coagulation in patients with a history of SARS-CoV-2 and probable bystander of thrombosis.

We also found out that the control and AMI groups without a history of SARS-CoV-2 had correlations of fibrinolysis parameters with components of the complement system or with acute-phase protein haptoglobin, respectively. It was previously shown in several studies that the fibrinolytic and complement systems are interrelated. On the one hand, plasmin is able to activate complement C3 and C5 [34,35,36]. On the other hand, it can inhibit activation of the complement system by cleaving complement factors [34]. Complement system components have also been shown to regulate fibrinolysis via different pathways. Mannan-binding lectin-associated serine protease 1 intensified fibrinolysis and, on the other hand, some complement components were shown to increase clot lysis time [37]. Other inflammatory mediators have also been found to be involved in the regulation of fibrinolysis [38]. In the work we report here, these correlations were absent in the control post-COVID and AMI post-COVID groups. It is possible that after SARS-CoV-2 infection the interrelation of the immune system with fibrinolysis is altered.

## 4. Limitations

Our study has a number of limitations. First, differences in the protein levels measured by LC-MS/MS with MRM are not equal to differences in the levels of functionally active proteins. Second, we did not consider the history of selective serotonin reuptake inhibitor usage that may possibly affect platelet aggregation results. Third, the use of the antiplatelet drugs during the prehospital stage in the AMI patients may influence the obtained results in the impedance aggregometry. Finally, the fact of COVID-19 was assessed retrospectively, which does not allow us to establish the exact date of COVID-19 infection accurately for each participant.

## 5. Materials and Methods

### 5.1. Subjects

The study was performed at I.V. Davydovsky Moscow City Clinical Hospital, from May to December 2022.

We included 66 patients with AMI and 58 healthy volunteers.

Inclusion criteria for the AMI group: confirmed diagnosis of AMI (ST-elevation myocardial infarction (STEMI) or non-ST-elevation myocardial infarction (non-STEMI)) according to the 4th universal definition of myocardial infarction (2018) [39].

Exclusion criteria for the AMI group: age > 90, acute SARS-CoV-2 infection on admission or less than 1 month before; unconfirmed diagnosis of AMI; active cancer; autoimmune disease; acute inflammation; anticoagulation therapy.

Inclusion criteria for healthy volunteers: no history of SARS-CoV-2 infection or no clinically significant symptoms after SARS-CoV-2 infection (post-COVID).

A validated questionnaire was used [40] to exclude the presence of post-COVID symptoms in healthy volunteers. The questionnaire assessed parameters such as dizziness/pre-syncope, syncope, orthostatic intolerance, palpitation/arrhythmia, dyspnea, fatigue, changes in blood pressure, depression, insomnia, drowsiness, memory loss, impaired concentration, nightmares, headache, weight loss, hair loss, loss of smell or taste. Volunteers having fewer than 5 points in all positions were included.

Exclusion criteria for healthy volunteers: age > 90 or <40, acute SARS-CoV-2 infection on admission or less than 1 month before, cardiovascular and pulmonary chronic diseases; diabetes mellitus; active cancer; autoimmune disease; acute inflammation; history of thromboembolism; treatment with antithrombotic drugs (some patients in the control group previously received antiplatelet drugs with no clear indications for them, all antiplatelet drugs were canceled the week before inclusion in the study).

The patients in each group were divided into two groups: post-COVID and control. The division of patients was performed in accordance with the level of IgG against N-protein (anti-N). We measured the titers of anti-N IgG in the serum using the automated indirect immunoassay ARCHITECT i1000SR analyzer with the compatible reagent kit (Abbott, Chicago, IL, USA), according to the manufacturer’s standard protocol. Patients were assigned to the control group if their anti-N IgG levels were less than 1.4. The post-COVID group comprised patients with higher anti-N IgG levels [41]. Finally, we had four groups: AMI post-COVID (*n* = 44), AMI control (*n* = 22), control post-COVID (*n* = 31), and control (*n* = 27).

### 5.2. Laboratory and Instrumental Analysis

All patients underwent physical examination, standard laboratory tests, including complete blood count (hemoglobin level, red blood cells (RBC), white blood cells (WBC), platelet count, etc.), a biochemical blood test (serum creatinine, alanine aminotransferase (ALT), aspartate aminotransferase (AST), etc.), and coagulation blood tests (D-dimer, activated partial thromboplastin time (APTT), prothrombin time (PT), international normalized ratio (INR), antithrombin III (AT-III), Quick prothrombin time test (PT Quick). The protocol is described in more detail in the Appendix A.

In addition to the standard blood testing, all patients underwent thrombodynamics, rotational thromboelastometry (in NATEM mode) to evaluate plasma coagulation, thrombodynamics in fibrinolysis mode to evaluate endogenous fibrinolysis, impedance aggregometry to evaluate platelet function, and plasma proteomics analysis.

Venous blood samples were obtained upon admission prior to percutaneous coronary intervention in the AMI group. The delay from venipuncture to hemostatic testing was less than 15 min. Whole blood was used for impedance aggregometry and rotational thromboelastometry, and platelet-free plasma was used for the thrombodynamics study. The protocol is described in more detail in the Appendix A.

### 5.3. Rotational Thromboelastometry

The study was performed in NATEM mode on the ROTEM (The Tem Innovations, GmbH, Germany). We assessed the following parameters: clotting time (CT, s), clot formation time (CFT, s), clot amplitudes at 10–30 min (A10–A30, mm), maximum clot firmness (MCF, mm), angle between the middle axis and the tangential line to the clotting curve through the 2 mm amplitude point (α, °), clot lysis index at 60 min (LI60%), and maximum lysis (ML,%). The reagents are described in more detail in the Appendix A.

### 5.4. Thrombodynamics

The study was conducted on a Thrombodynamics Analyzer System T-2 (HemaCore LLC, Moscow, Russia). The following parameters of clot growth were assessed: clot growth rate (V, μm/min), initial and stationary clot growth rate (Vi, μm/min; Vst, μm/min), lag time, the delay between the test start and the clot formation onset (Tlag, min), clot size (CS, μm), clot density (D, arb units), and spontaneous clots formation time (Tsp, min). An activator with urokinase was added to induce thrombus lysis and assess the fibrinolysis parameters, such as lysis onset time (LOT, min), the rate of lysis progression (LP, %/min), the clot lysis time (CLT, min), percent of remaining clot density (LI, %), and the expected clot lysis time (LTE, min) [42]. The reagents are described in more detail in the Appendix A.

### 5.5. Impedance Aggregometry

A Multiplate analyzer (Roche Diagnostics International Ltd., Rotkreuz, Switzerland) was used to assess impedance aggregometry. We used arachidonic acid (ASPI), adenosine diphosphate (ADP), and thrombin receptor-activated peptide-6 (TRAP-6) for platelet activation. The test time was 6 min. Platelet aggregation was assessed from the area under the curve (AUC). The reagents are described in more detail in the Appendix A.

### 5.6. Targeted Plasma Proteomics

We carried out targeted proteomic analysis using liquid chromatography–tandem mass spectrometry (LC-MS/MS) with multiple-reaction monitoring (MRM). We used synthetic stable isotope-labeled internal standard (SIS) and natural (NAT) synthetic proteotypic peptides for measuring the corresponding 227 proteins in plasma. The selected SIS and NAT synthetic peptides had been previously validated for use in LC/MRM-MS experiments for blood plasma [12]. Isotopically labeled peptide standards were synthesized at Skoltech within the framework of the Megagrant of the Ministry of Science and Education of Russia, “Next generation proteomics for improvement of personalized medicine and health” (Agreement # 075-10-2022-090). We generated standard curves using NAT and SIS peptide standards, with bovine serum albumin (BSA) as a surrogate matrix. All samples were analyzed in duplicate with high-performance liquid chromatography mass spectrometry (HPLC-MS), using an ExionLC™ UHPLC system coupled online to a SCIEX QTRAP 6500+ triple-quadrupole mass spectrometer (SCIEX, Toronto, ON, Canada). We carried out HPLC separation using Zorbax Eclipse Plus RRHD C18 RP-UHPLC (150 × 2.1 mm, i.d., 1.8 μm particles; Agilent Technologies, Santa Clara, CA, USA) with gradient elution. We carried out mass-spectrometric measurements using the MRM acquisition method. The corresponding transition list for MRM experiments with Q1; Q3 masses for each peptide is available in Appendix A. For quantitative analysis of LC-MS/MS raw data, we used the Skyline Quantitative Analysis software (version 20.2.0.343, University of Washington, Washington, DC, USA). The protocol is described in more detail in the Appendix A.

The MRM data quality was checked manually in Skyline for all selected proteins/peptides, and includes the absence of interference peaks and the good quality of the peak shape, and the ratios of the precursor and product ion. The exemplary MRM data (from Skyline) for selected proteins are presented in Appendix A. All experimental results from MRM analysis were uploaded to the PeptideAtlas SRM Experiment Library (PASSEL) and are available via the link: http://www.peptideatlas.org/PASS/PASS04817 (accessed on 15 March 2023).

### 5.7. Statistical Analysis

Statistical analysis was performed with R (4.0.5). The expression values obtained in the present study were in not normally distributed in most cases, according to the Shapiro–Wilk test, and therefore are represented as medians and interquartile ranges [Q1; Q3]. For comparison of several groups, we used the Mann–Whitney rank test, with a continuity correction. For the analysis of categorical parameters, we used a two-tailed Fisher’s exact test with 2 × 2 frequency tables, and a chi-square test of independence for features that included more than two categories. In order to overcome errors from multiple comparisons, we performed a Benjamini–Hochberg FDR correction with calculation of critical values for each comparison matched with corresponding *p*-values; we calculated adjusted *p*-values and compared them with a critical value of 0.1 to keep the positive false discovery rate below 10%. For calculation of the Spearman’s coefficient (rho), we used a threshold of *p* adjusted < 0.05. To test for differences in partial correlations between the groups within each interval, the partial correlations were first converted into an intermediary statistic, using Fisher’s z transformation. We inferred significant differences in partial correlations on the basis of a *p* adjusted < 0.05 and differences in the magnitude of correlation of at least 0.4 and rho > 0.5. We constructed a network of differential correlations for each group by linking/connecting the parameter with significant differences in partial correlations [43]. We performed clusterization using the k-medoids algorithm and a correlation matrix used as a matrix of distance. For defining the cluster number, we used gap statistics via bootstrapping with Monte Carlo simulation and different centroids from cluster package (version 2.1.4) [44]. The 10 initial random centroids were initially configured, and 300 bootstraps of Monte Carlo were generated.

In 2.1% of the subjects from whom proteomics data were obtained, and 6.7% of the subjects from whom laboratory test data were obtained, there were no values for various parameters, so we used the pairwise deletion approach to calculate the correlation coefficients. The available data sets are sufficient for demonstration of significant differences between compared groups using the Mann–Whitney rank test, sig. level = 0.1, power = 0.8, and effect size = 0.9 (which corresponds to ‘large’ effect size) and using a Spearman’s rank correlation, sig. level = 0.05, power = 0.8, and r = 0.6, based on the smallest compared group [45].

## 6. Conclusions

Our study demonstrated the elevation of inflammatory protein levels and activation of coagulation in AMI, which corresponded to previous data. After SARS-CoV-2 infection, AMI patients had less noticeable growth of acute-phase proteins and markers of hemostasis, compared with AMI patients without SARS-CoV-2 infection, that could be explained by prolonged immune system alteration after COVID-19. This fact is a new finding and requires further research.

## Figures and Tables

**Figure 1 ijms-24-06523-f001:**
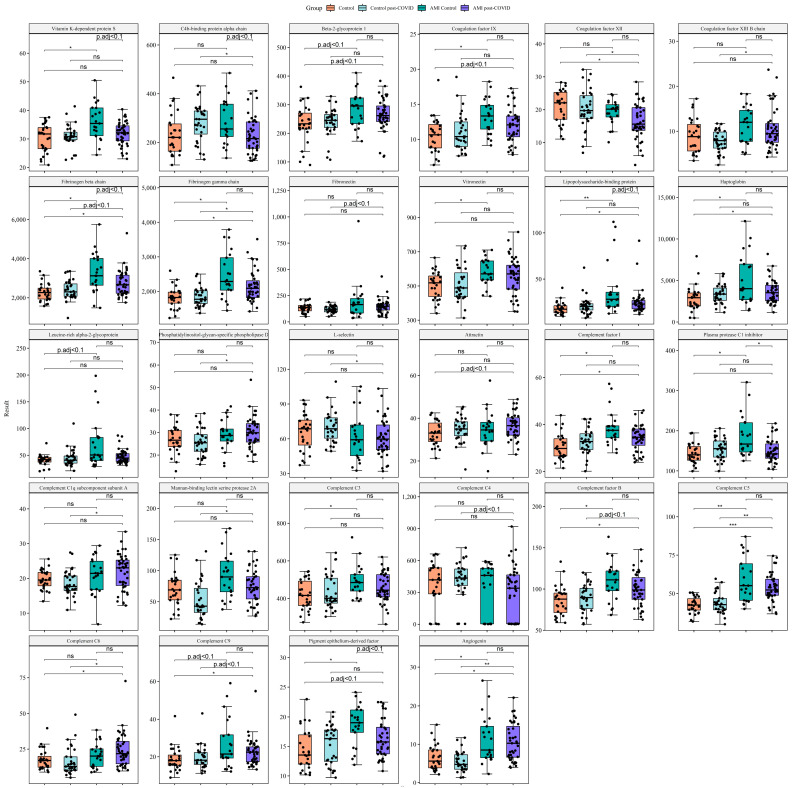
Main comparisons of proteomics data between study groups. Targeted proteomic analysis was carried out using liquid chromatography–tandem mass spectrometry (LC-MS/MS) with multiple-reaction monitoring (MRM). For comparison of several groups, we used the Mann–Whitney rank test with a continuity correction. In order to overcome errors from multiple comparisons, we performed a Benjamini–Hochberg FDR correction, with calculation of critical values for each comparison matched with corresponding *p*-values; we calculated adjusted *p*-values and compared them with a critical value of 0.1 to keep the positive false discovery rate below 10%. Data are presented as the median [Q1; Q3]. * *p* adj. < 0.05, ** *p* adj. < 0.01, *** *p* adj. < 0.001, ns, not significant. The boxplots include proteins different between the AMI control and AMI post-COVID groups: C4b-binding protein alpha-chain, fibrinogen beta-chain, lipopolysaccharide-binding protein, PEDF, plasma protease C1 inhibitor, vitamin-K-dependent protein S. Based on their function, we focused on proteins connected to hemostasis, acute-phase proteins, and components of the complement system. In the boxplots, we included proteins with these functions if they were different in at least one of the comparisons: AMI control vs. control, AMI post-COVID vs. control, AMI post-COVID vs. control post-COVID.

**Figure 2 ijms-24-06523-f002:**
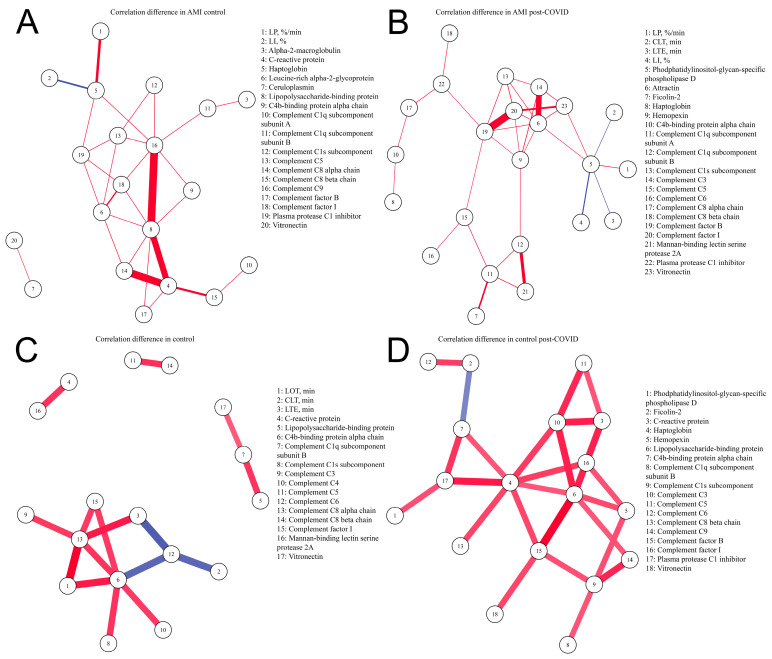
Differences in correlations between parameters of fibrinolysis, proteins involved in inflammation and components of the complement system. Differences in partial correlations were inferred on the basis of *p* adj. < 0.05, a magnitude of a Spearman’s correlation coefficient of at least 0.4, and a Spearman’s correlation coefficient > 0.5 or <−0.5. In each group, we calculated the correlation coefficients between the parameters, and then compared the obtained coefficients between the groups. If there is a statistically significant correlation (*p* adj. < 0.05) between the parameters in both groups, and if the correlation coefficients differ by more than 0.4, then such a correlation will be displayed in the figure between the correlating parameters. Red—positive correlation, blue—negative correlation. Line thickness depends on the Spearman’s correlation coefficient. Parameters of thrombodynamics: LOT—lysis onset time (min), LP—the rate of lysis progression (%/min), CLT—the clot lysis time (min), LI—percent of remaining clot density (%), LTE—the expected clot lysis time (min). (**A**) Correlations present in the AMI control group and absent in the AMI post-COVID group; (**B**) correlations present in the AMI post-COVID group and absent in the AMI control group; (**C**) correlations present in the control group and absent in the control post-COVID group; (**D**) correlations present in control post-COVID group and absent in the control group.

**Figure 3 ijms-24-06523-f003:**
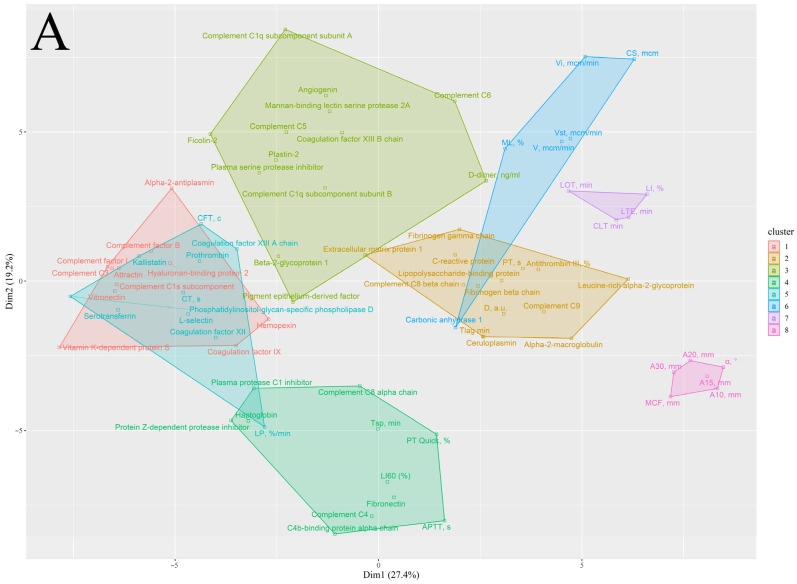
Clusterization of parameters of hemostasis, and proteins involved in inflammation and hemostasis, components of the complement system, and two proteins affecting the endothelium (angiogenin, PEDF). We performed clusterization using the k-medoids algorithm and a correlation matrix used as a matrix of distance. For defining the number of clusters, we used gap statistics via bootstrapping with Monte Carlo simulation and different centroids. Parameters of coagulation blood tests: APTT—activated partial thromboplastin time (s), PT—prothrombin time (s), PT Quick—Quick prothrombin time test (%). Parameters of rotational thromboelastometry: CT—clotting time (sec), CFT—clot formation time (s), A10–A30—clot amplitudes at 10–30 min (mm), MCF—maximum clot firmness (mm), α—angle between the middle axis and the tangential line to the clotting curve through the 2 mm amplitude point (°), LI60—clot lysis index at 60 min (%), ML—maximum lysis (%). Parameters of thrombodynamics: V—clot growth rate (μm/min), Vi—initial clot growth rate (μm/min), Vst—stationary clot growth rate (μm/min), Tlag—lag time, the delay between the test start and the clot formation onset (min), CS—clot size (μm), D—clot density (arb units), Tsp—spontaneous clots formation time (min), LOT—lysis onset time (min), LP—the rate of lysis progression (%/min), CLT—the clot lysis time (min), LI—percent of remaining clot density (%), LTE—the expected clot lysis time (min). (**A**) AMI post-COVID group; (**B**) AMI control group.

**Table 1 ijms-24-06523-t001:** Comparison of clinical data between study groups. BMI—body mass index, NSAID—non-steroidal anti-inflammatory drugs. Green: *p* adj. < 0.1.

	AMI Control	AMI Post-COVID	Control	Control Post-COVID	*p* adj. AMI Control vs. AMI Post-COVID	*p* adj. AMI Control vs. Control	*p* adj. AMI Post-COVID vs. Control Post-COVID	*p* adj. AMI Post-COVID vs. Control
Male sex, *n* (%)	22, (73%)	44, (84%)	27, (37%)	31, (26%)	1	1	<0.01	0.01
Age, years Med [Q1; Q3]	65.5 [56.25; 71.5]	59 [50.75; 72]	48 [43.5; 54.5]	49 [47; 55.5]	0.29	<0.01	<0.01	<0.01
BMI Med [Q1; Q3]	27.91 [24.9; 30.83]	26.73 [24.39; 30.9]	24.38 [23.13; 26.73]	26.4 [22.87; 29.66]	0.82	0.05	0.55	0.04
Smoking, *n* (%)	18, (44%)	37, (65%)	25, (16%)	31, (23%)	1	1	0.09	0.03
Arterial hypertension, *n* (%)	22, (100%)	44, (86%)	25, (32%)	31, (32%)	1	<0.01	<0.01	<0.01
Diabetes mellitus, *n*, (%)	22, (27%)	44, (16%)	25, (0%)	31, (0%)	1	0.96	1	1
History of MI, *n* (%)	22, (23%)	44, (14%)	25, (0%)	31, (0%)	1	1	1	1
Chronic obstructive pulmonary disease, *n* (%)	22, (5%)	44, (2%)	25, (0%)	31, (0%)	1	1	1	1
Chronic kidney disease, *n* (%)	22, (9%)	43, (7%)	25, (0%)	31, (3%)	1	1	1	1
History of PCI, *n* (%)	22, (18%)	44, (9%)	25, (0%)	31, (0%)	1	1	1	1
Antiplatelet drugs, *n* (%)	22, (23%)	41, (17%)	25 *, (12%)	31 *, (3%)	1	1	1	1
Statins, *n* (%)	22, (27%)	41, (7%)	25, (20%)	31, (3%)	1	1	1	1
ACE inhibitors/AT-II blockers, *n* (%)	21, (62%)	40, (30%)	25, (20%)	31, (16%)	1	0.84	1	1
NSAIDs	19, (0%)	40, (0%)	27, (0%)	31, (0%)	1	1	1	1
SARS-CoV-2 Vaccination, *n* (%)	22, (55%)	43, (53%)	27, (63%)	31, (81%)	1	1	1	1
History of bleeding, *n* (%)	22, (5%)	43, (9%)	25, (4%)	31, (0%)	1	1	1	1

* All antiplatelet drugs were canceled the week before the inclusion in the control groups (see exclusion criteria).

**Table 2 ijms-24-06523-t002:** Comparison of laboratory data between study groups. LDL—low-density lipoproteins, HDL—high-density lipoproteins, APTT—activated partial thromboplastin time, PT Quick—Quick prothrombin time test, PT—prothrombin time, INR—international normalized ratio, ALT—alanine aminotransferase, AST—aspartate aminotransferase, AT-III—antithrombin III. Green: *p* adj. < 0.1. Data are presented as the median [Q1; Q3].

	AMI Control	AMI Post-COVID	Control	Control Post-COVID	*p* adj. AMI Control vs. AMI Post-COVID	*p* adj. AMI Control vs. Control	*p* adj. AMI Post-COVID vs. Control Post-COVID	*p* adj. AMI Post-COVID vs. Control
Hemoglobin, g/L	143.5 [132.7; 162]	150 [131; 155.2]	133 [122.5; 142.5]	137 [132.5; 140.5]	0.542	0.078	0.172	0.084
Platelets, ×10^9^/L	272.5 [228; 317.5]	237.5 [194; 282]	254 [203; 295]	243 [208.5; 273]	0.382	0.702	0.611	0.619
Leukocytes, ×10^9^/L	9.96 [8.36; 11.68]	10.34 [8.95; 12.1]	5.39 [4.7; 5.99]	5.25 [4.82; 6.4]	0.84	<0.01	<0.01	<0.01
Total cholesterol, mmol/L	4.94 [4.08; 6.39]	5.6 [4.89; 6.35]	5.07 [4.57; 5.98]	5.65 [4.91; 6.04]	0.388	0.77	0.92	0.376
LDL, mmol/L	3.01 [2.3; 4.25]	3.48 [3.01; 4.08]	2.89 [2.28; 3.7]	3.1 [2.64; 3.46]	0.715	0.668	0.119	0.152
HDL, mmol/L	1.1 [0.87; 1.19]	1.07 [0.91; 1.29]	1.67 [1.34; 2.02]	1.79 [1.61; 2.36]	0.909	<0.01	<0.01	<0.01
Triglycerides, mmol/L	1.85 [1.22; 2.28]	1.72 [1.15; 2.34]	1.02 [0.82; 1.65]	1.02 [0.82; 1.2]	0.889	0.013	<0.01	0.013
Creatinine, µmol/L	102 [91.2; 124.7]	86.5 [79; 106.5]	73 [63; 90.5]	74 [66; 86.5]	0.128	<0.01	<0.01	0.013
APTT, s	25.95 [21.72; 30.42]	27.8 [24.3; 30.65]	30.4 [29.05; 32.75]	32.9 [30.7; 34.6]	0.58	0.036	<0.01	0.036
PT Quick, %	86.7 [76.9; 102.05]	90.3 [77.75; 98.3]	106 [101; 110.5]	102 [97; 112.5]	0.804	<0.01	<0.01	<0.01
PT, s	10.95 [10.6; 12.25]	11.4 [10.6; 11.83]	11.1 [10.85; 11.6]	11.4 [10.75; 11.9]	0.801	0.752	0.95	0.84
INR	1.08 [0.99; 1.15]	1.06 [0.99; 1.15]	0.94 [0.92; 0.98]	0.97 [0.91; 1]	0.711	<0.01	<0.01	<0.01
D-dimer, ng/mL	309 [181.5; 369]	282.5 [187.25; 426]	285 [211.5; 473]	287 [194; 428]	0.979	0.954	0.94	0.92
ALT, IU/L	25.5 [19.2; 36.7]	29.5 [21.5; 39]	14 [12; 24]	17 [12.5; 24]	0.768	0.034	<0.01	<0.01
AST, IU/L	33.5 [24.5; 58.25]	27.5 [21; 49.25]	19 [16; 21]	18 [16.5; 23.5]	0.649	<0.01	<0.01	<0.01
AT-III, %	92.5 [89; 101]	91 [88.75; 96.25]	101 [96; 111.0]	100 [95; 107]	0.663	0.057	<0.01	<0.01

## Data Availability

All experimental results from MRM analysis were uploaded to the PeptideAtlas SRM Experiment Library (PASSEL) and are available via the link: http://www.peptideatlas.org/PASS/PASS04817 (accessed on 15 March 2023).

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
