# Peer review of "Targeted Blood Plasma Proteomics and Hemostasis Assessment of Post COVID-19 Patients with Acute Myocardial Infarction"

_ijms, 2023, doi:10.3390/ijms24076523_

Round 1

Reviewer 1 Report

The reviewed manuscript “Targeted blood plasma proteomics and hemostasis assessment of post COVID-19 patients with acute myocardial infarction” by Anna Kalinskaya et al. concerns an important issue of possible cardiac complications related to the coronavirus infection. The correlation of physiological parameters is a relatively frequent subject of research in this context. However, the use of targeted plasma proteomics has a high potential for discovering new relationships and potential therapeutic or diagnostic applications. The study uses a classic panel of diagnostic tests in acute myocardial infarction, but some doubts are raised by an attempt to correlate it with proteomic tests. Targeted proteomics is a state-of-the-art tool that enables the precise determination of protein levels based on representative peptides in a large number of samples.

Testing the level of a peptide representing a given protein is not synonymous with finding a change in the activity of a functional protein in the serum. The presence of peptides may be the result of proteolytic degradation. What were the criteria for selecting the panel of tested proteins. Isn't the assumption of greater importance of certain processes a limitation? Wouldn't it be better to use a more global proteomic approach (e.g. iTRAQ, label free, SWAT) when studying new relationships? The manuscript lacks a brief justification for the use of the method, what are its advantages in this case.

The captions to the figures are insufficient and imprecise. They do not provide the reader with sufficient information to properly understand the research results.

fig1. Main comparisons of proteomics data between study groups.

fig2. Main correlations between proteomics data and parameters of hemostasis.

fig3. Differences in correlations between parameters of ...

Line 149 - "The main results of proteomics data comparisons between study groups are presented in Fig 1." - what are the protein selection criteria for fig1. What is the selection of factors on the diagonal and horizontally at the top (fig.2). Figure 3 is incomprehensible to me. What does red and what blue mean? How was it generated? What is the difference between the nodes shown twice in the graphs e.g. Fig3B no 7 and no 21 or Fig3D no 2 and 17; what are the conclusions from this figure?

How fig 4 was generated, please provide details?

References are not in the correct format for the journal.

Validation of at least some results of proteomic analysis (e.g. conclusion on acute phase proteins - ELISA or immunoblot) would increase the value of the work.

Proteomics data have not been deposited in a proper repository.

Author Response

Question 1: Testing the level of a peptide representing a given protein is not synonymous with finding a change in the activity of a functional protein in the serum. The presence of peptides may be the result of proteolytic degradation. What were the criteria for selecting the panel of tested proteins. Isn't the assumption of greater importance of certain processes a limitation? Wouldn't it be better to use a more global proteomic approach (e.g. iTRAQ, label free, SWAT) when studying new relationships? The manuscript lacks a brief justification for the use of the method, what are its advantages in this case.

Answer 1:

Thank you for your useful comment! The use of targeted proteomics with validated assay of peptide standards and robust instrumentation such as triple quadrupole provides the results which have more potential for further translation in clinical practice. Moreover, the targeted approach allows the production of consistent results across different instrumental platforms and thus allows biomarker validation using the exact same methods and workflows in independent cohorts and laboratories (https://doi.org/10.1021/acs.analchem.1c03782, https://doi.org/10.1007/978-1-0716-1024-4_25).

In the current study, we considered the proteins and corresponding peptides panel from the BAK270 MRM assay (https://doi.org/10.1039/C9AN01893J). The assay is developed for analysis of potential protein biomarkers for cardiovascular disease including 61 FDA approved biomarkers in blood plasma. Moreover, the robustness of current MRM assay for selected blood proteins was recently demonstrated (https://doi.org/10.1016/j.mcpro.2022.100212).

We also agree that it is always important to evaluate and validate the peptides which are selected for protein quantitation. In our study, we selected the proteomic strategy based on LC-MS/MS and multiple reaction monitoring (MRM) with stable isotope labeled peptide standards (SIS) which is a gold standard for quantitative proteomic analysis and precision medical diagnostics. The important issue is that we utilize a BAK270 MRM assay previously developed and validated by Prof. Borchers group for blood plasma protein analysis (https://doi.org/10.1039/C9AN01893J). With MRM, there are at least 3 levels of specificity for each selected protein/peptide transitions:  1st - the masses of the precursor (Q1), 2nd - the product ions (Q2) and 3rd - the retention time (RT). The information from previous proteomics validation and experimental work, which were performed in various laboratories across the globe, is hosted in our knowledgebase MRMAssayDB  (http://mrmassaydb.proteincentre.com).

We agree that the question regarding proteolytic degradation is important. And in our case this is another important advantage of current MRM based strategy if compared with immunoassay methods. The robustness of current assay and MRM approach was recently demonstrated by Prof. Borchers group (https://doi.org/10.1016/j.mcpro.2022.100212). We thus may conclude that for plasma proteomic analysis for selected proteins no protease inhibitors are needed. But we acknowledge this question from the reviewer and agree that this issue should be always checked for each protein selected for analysis. The corresponding text was added to the Introduction section of the manuscript (page 1, line 46).

Question 2: The captions to the figures are insufficient and imprecise. They do not provide the reader with sufficient information to properly understand the research results.

fig1. Main comparisons of proteomics data between study groups.

fig2. Main correlations between proteomics data and parameters of hemostasis.

fig3. Differences in correlations between parameters of ...

Answer 2: According to your suggestion we have updated the captions to the figures in the main text of the manuscript (Figures 1-3) and in the supplemental (Figures S4, S5). Figure 2 was moved to the supplemental and renamed in Figure S4.

Question 3: Line 149 - "The main results of proteomics data comparisons between study groups are presented in Fig 1." - what are the protein selection criteria for fig1. What is the selection of factors on the diagonal and horizontally at the top (fig.2). Figure 3 is incomprehensible to me. What does red and what blue mean? How was it generated? What is the difference between the nodes shown twice in the graphs e.g. Fig3B no 7 and no 21 or Fig3D no 2 and 17; what are the conclusions from this figure?

Answer 3: Thank you for your comment. We updated the information in the figure’s captions and in the main text of the manuscript (page 6, line 162; page 7, line 181; page 8, line 206; page 9, line 223).

Figure 1: First, we included in the boxplots all the proteins different between AMI control and AMI post-COVID groups: C4b-binding protein alpha chain, fibrinogen beta chain, lipopolysaccharide-binding protein, pigment epithelium-derived factor (PEDF), plasma protease C1 inhibitor, vitamin K-dependent protein S. Based on their function, we decided to further focus on proteins connected to hemostasis, acute-phase proteins and components of the complement system. We included in the boxplots proteins with these functions if they were different in at least one of the comparisons: AMI control vs. control, AMI post-COVID vs. control, AMI post-COVID vs. control post-COVID. One of the main functions of PEDF is negative regulation of angiogenesis, therefore we also added a protein with angiogenic function which was also different in the comparisons mentioned above (angiogenin).

Figure 2 (Figure S4 in the revised version of the manuscript): We visualized parameters of hemostasis and proteins connected to hemostasis if they correlated with proteins connected to inflammation (our division of proteins into functional groups is presented in Table S3) and components of the complement system in at least one study group. And vice versa: we included proteins connected to inflammation and components of the complement system if they correlated with parameters of hemostasis and proteins connected to hemostasis in at least one study group. This was done to make the correlation matrices easy to read and not congested. The correlation matrices are symmetric and therefore the parameters mentioned at the top and on the diagonal of the matrices are the same.

Correlation matrices were made based on the correlation tables S6-S9. Full correlation tables include all the 81 proteins and all the parameters of hemostasis analyzed in the study, and p adjustment was done for this list of correlations.

Red – positive correlations, blue – negative correlations.

Figure 3 (Figure 2 in the revised version of the manuscript): This figure was generated based on the correlation tables (Tables S4, S5, S6, S7 - now renamed into S6, S7, S8, S9, because Table S2 was divided into Tables S2 and S4 containing parameters of hemostasis and results of proteomics, respectively, and Table S5 was added).

Initially, in each group, we calculated the correlation coefficients between the parameters, and then compared the obtained coefficients between the groups. If there is a statistically significant correlation (p adj<0.05) between the parameters in both groups, and if the correlation coefficients differ by more than 0.4, then such a correlation will be displayed in the figure between the correlating parameters. The essence of these figures is to see the difference in correlations of the same parameters between the compared groups. Red - positive correlation, blue - negative correlation. Line thickness depends on the Spearman’s correlation coefficient.

In this figure, we included only parameters of fibrinolysis, proteins connected to inflammation and components of the complement system to illustrate that parameters of fibrinolysis correlated with components of the complement system or haptoglobin in the control and AMI control groups and did not correlate with any acute-phase proteins or components of the complement system in control post-COVID and AMI post-COVID groups.

Thank you very much! We deleted duplicate parameters in Fig3B and Fig3D.

Question 4: How fig 4 was generated, please provide details?

Answer 4: Thank you for your comment! We provided the details to Figure 4 (Figure 3 in the revised version of the manuscript) (page 11, line 272). Parameters used for clusterization included parameters of thrombodynamics and rotational thromboelastometry, results of coagulation blood tests, proteins involved in inflammation and hemostasis, components of the complement system, and of two proteins affecting endothelium (angiogenin, PEDF).

We performed clusterization using the k-medoids algorithm and a correlation matrix used as a matrix of distance. For defining the number of clusters, we used gap-statistics via bootstrapping with Monte-Carlo simulation and different centroids from R, cluster package (version 2.1.4) (Peter Rousseeuw; Anja Struyf; Mia Hubert; based on Kaufman and Rousseeuw Methods for Cluster Analysis; “Finding Groups in Data,” 1990). In this picture, first of all, the mutual arrangement of parameters relative to each other is important, and their clustering occurs on the basis of correlation matrix.

Question 5: References are not in the correct format for the journal.

Answer 5: Thank you for your comment! We changed the format of the references.

Question 6: Validation of at least some results of proteomic analysis (e.g. conclusion on acute phase proteins - ELISA or immunoblot) would increase the value of the work.

Answer 6:

Thank you for your question! We agree that the validation of our findings would be necessary in the future for the most interesting and prospective proteins. To increase the value of this work we added the correlations between C-reactive protein proteomics and wide range C-reactive protein, measured by immunoturbidimetry method on Siemens ADVIA  2400 Chemistry Analyzer (Siemens Healthcare Diagnostics Inc, USA) and between fibrinogen beta chain proteomics and Clauss fibrinogen measured by photo-optical detection method on ACL TOP 300 CTS (Instrumentation Laboratory, USA) (Figure S6, page 6, line 177 in the main text of the manuscript).

The collection of samples from alternative patient cohorts has already started. We have also ordered the antibodies for the immunoassays; it takes 3-4 months till delivery.

Question 7: Proteomics data have not been deposited in a proper repository.

Answer 7:

Thank you for this remark! All experimental results from MRM analysis were uploaded to the PeptideAtlas SRM Experiment Library (PASSEL) and are available via link:

http://www.peptideatlas.org/PASS/PASS04817

The corresponding information was added to the manuscript (page 15, line 467; page 19, line 695).

Reviewer 2 Report

The authors study molecular mechanisms underlying cardiovascular complications after the SARS-CoV-2 infection. The goal of the study was to analyze the features of blood coagulation, platelet aggregation, and plasma proteomics in COVID-19 convalescents with acute myocardial infarction. They used targeted proteomic analysis with liquid chromatography-tandem mass spectrometry (LC-MS/MS) with multiple reaction monitoring (MRM) and synthetic stable-isotope labeled internal standard and natural synthetic proteotypic peptides for measuring the corresponding 227 proteins in plasma.

This is a well-written paper and carefully executed study.

I have the following comments:

The authors used gender-mixed groups of probands with an excess of males in some groups. It is clear that it is not easy to assemble proper study cohorts, but the authors should at least discuss and not only mention their limitations. The differences in age, BMI, gender and so forth may influence the final results. Moreover, gender, in particular, should be tested, because the female cycle causes major differences in the proteome.

The authors use a method, which has been published by other authors. They should at least demonstrate that the experiment works in their setting as well by showing exemplary data. I know from experience that Skyline can suggest false values from overlayed or weak data, even in MRM. The authors should discuss their means of data quality control.

Why did the authors discuss non-significant data? They certainly can do that but then they should explain their reasoning (p 4-5, Table S2).   

Table S2: p is typically < 0.05. Why did they mark p > 0.05?

Use journal reference format

Author Response

Question 1: The authors used gender-mixed groups of probands with an excess of males in some groups. It is clear that it is not easy to assemble proper study cohorts, but the authors should at least discuss and not only mention their limitations. The differences in age, BMI, gender and so forth may influence the final results. Moreover, gender, in particular, should be tested, because the female cycle causes major differences in the proteome.

Answer 1: Thank you for your comment! We tested the influence of gender, age and BMI on the results obtained. It turned out that age and gender had no noticeable influence on the results (Spearman’s correlation coefficient>0.5 or <-0.5, p adj<0.05) (See Table S5, page 6, line 172 in the main text of the manuscript). BMI correlated with the complement С3, complement factor I, and complement factor B levels (Spearman’s correlation coefficient>0.5, p adj<0.05). We added this fact to the study discussion section of the manuscript (page 11, line 295).

Question 2: The authors use a method, which has been published by other authors. They should at least demonstrate that the experiment works in their setting as well by showing exemplary data. I know from experience that Skyline can suggest false values from overlayed or weak data, even in MRM. The authors should discuss their means of data quality control.

Answer 2:

Thank you for this remark! In our study, we selected the proteomic strategy based on LC-MS/MS and multiple reaction monitoring (MRM) with stable isotope labeled peptide standards (SIS) which is a gold standard for quantitative proteomic analysis and precision medical diagnostics. The important issue is that we utilize a BAK270 MRM assay previously developed and validated by Prof. Borchers group for blood plasma protein analysis (https://doi.org/10.1039/C9AN01893J). With MRM, there are at least 3 levels of specificity for each selected protein/peptide transitions:  1st- the masses of the precursor (Q1), 2nd - the product ions (Q2) and 3rd - the retention time (RT). The information from previous proteomics validation and experimental work, which were performed in various laboratories across the globe, is hosted in our knowledgebase MRMAssayDB  (http://mrmassaydb.proteincentre.com ).

We agree that it is always important to check the data quality according to the SRM/MRM guidelines which includes for example the absence of interference peaks and the good quality of the peak shape, and the ratios of the precursor and product ion.

In the current study we additionally checked all selected proteins manually. As recommended the exemplary data were added to the Supplementary file as supporting data for selected proteins (Figures S1-S3).

Additionally, all experimental results from MRM analysis were uploaded to the PeptideAtlas SRM Experiment Library (PASSEL) and are available via link:

http://www.peptideatlas.org/PASS/PASS04817

This information was added to the manuscript (page 15, line 465) and to the Supplementary file.

Question 3: Why did the authors discuss non-significant data? They certainly can do that but then they should explain their reasoning (p 4-5, Table S2).  

Question 4: Table S2: p is typically < 0.05. Why did they mark p > 0.05?

Answer 3,4: Thank you for your comment! Our study included 500 comparisons of numeric parameters (125 numeric parameters multiplied by 4 comparisons: AMI control vs. AMI post-COVID, AMI control vs. control, AMI post-COVID vs. control, AMI post-COVID vs. control post-COVID). In order to overcome errors from multiple comparisons we performed a Benjamini-Hochberg FDR correction with calculation of critical values for each comparison matched with corresponding p-values; we calculated adjusted p-values and compared them with a critical value of 0.1 to keep the positive false discovery rate below 10% (page 15, line 478 of the manuscript). Threshold of the p adjusted was raised to 0.1 due to the big number of comparisons. This statistical method is often used in case of multiple comparisons (https://doi.org/10.1111/j.1538-7836.2009.03522.x, https://doi.org/10.1016/j.thromres.2011.02.012).

Question 5: Use journal reference format

Answer 5: We changed the format of the references.

Reviewer 3 Report

In this manuscript, Kalinskaya et al study the profiles of various hemostasis parameters in 66 patients with acute myocardial infection according to their COVID-19 serological status and compare them to 58 healthy subjects. By doing this, they claim to better understand the pathophysiological pathways involved in cardiovasculat complications following COVID-19 infection.

Although the topic is seducing and the quantity of results is significant, the methodological approach prensents serious flaws and the presentation of results lacks clarity and soundness.

1/The major point is the retrospective design of the study: the history of COVID-19 infection has been determined on serological results, so the precise time of infection is completely unknown (can be from 1 months to 2 years!) A prospective design with longitudinal follow-up of biomarkers would have been more appropriate (at least on healthy volunteers to give an idea of the evolution of those biomarkers following COVID-19 infection)

2/Healthy volunteers with a known history of COVID-19 infection were exluded wich means that controls post-COVID were all asymptomatic at the time of COVID-19 infection. This is not the case for AMI patients. Why ? Biomarkers can potentially vary with the severity of symptoms at the time of infection...

3/I could not find any mention of ethical approval and informed consent from the patients throughout the manuscript.

4/Table 1: 12% and 3% of controls received antiplatelet drugs at the time of inclusion which is surprising seems a lot. Can the authors comment about this? 

5/Patients under anticoagulant drugs were excluded, while those under antiplatelet drugs were included. However, antiplatelet drugs do modify some hemostasis parameters, especially impedance aggregometry. Was this considered in the statistical analysis ?

6/Were patients/controls free of NSAIDs and SSRIs ?

7/Methods to measure standard laboratory parameters should be described: reagents and analysers ?

8/I am suprised by the levels or D-dimers that remain completely normal even in the AMI groups. Could the authors comment on this ?

9/Table S2: For more clarity, results from TEG, thrombodynamics and aggregometry should be presented separately from proteomics.

10/Level of protein S and C4BP are both decreased in AMI-post COVID patients and are positively correlated. The authors thus hypothesize that decreased protein S could be related to the occurence of thrombosis in these patients. This is completely overinterpreted, as protein S measured by proteomics does not reflect its anticoagulant activity and it is well known that C4BP decreases PS activity without changing its antigen level.

11/Same remark about coagulation factors measured by proteomics: results should be interpreted with caution as activity might differ from antigen levels...

12/Some parameters presented in Table S1 are not in Table S2. How were parameters chosen ?? Other coagulation factors such as FVIII would have been of great interest..

Minor points:

1/Abstract line 27: acute-phase proteins do not grow...

2/Page 1 line 42: the search of biomarkers to stratify cardiovascular risks after COVID-19 infection  would have required a propective design (cf supra)

3/D-dimer results are both in Table 2 and Table S2.

4/Page 15, line 328: antithrombotic drugs include antiplatelet agents. According to Table 1, some healthy volunteers received antiplatelet drugs. Please clarify.

4/Correlations between biomarkers in the different groups are of no interest and are difficult to interprete (they are not for most of them...) So all the correlation matrices should be moved in the supplemental material.

Author Response

Question 1: The major point is the retrospective design of the study: the history of COVID-19 infection has been determined on serological results, so the precise time of infection is completely unknown (can be from 1 months to 2 years!) A prospective design with longitudinal follow-up of biomarkers would have been more appropriate (at least on healthy volunteers to give an idea of the evolution of those biomarkers following COVID-19 infection)

Answer 1: Thank you very much for your question! The retrospective nature of the study was chosen due to the extreme complexity of constructing a prospective study. Since the prevalence of myocardial infarction in the studied geographic region is about 2.9% (https://doi.org/10.15829/1560-4071-2022-4952), the recruitment of the number of patients with myocardial infarction presented in our study (n=66) requires a prospective enrollment of 2275 participants, which is associated with a large number of technical and organizational difficulties.

Each participant included in the study had a carefully collected anamnesis regarding the past COVID-19 infection. However, a significant part of the participants underwent only a rapid test during the illness, which is hard to confirm afterwards. An increase in anti-N IgG antibodies is a widespread and objective marker of a past coronavirus infection. According to the literature, anti-N IgG antibodies persist for a long time (12-16 months) after COVID-19 disease (https://doi.org/10.1038/s41598-022-16402-0; https://doi.org/10.1038/s41598-021-02659-4) and, therefore, can be used as a convenient marker to separate study participants.

Thank you for the great suggestion about healthy volunteers. We will do it for sure in our future study.

Question 2: Healthy volunteers with a known history of COVID-19 infection were excluded wich means that controls post-COVID were all asymptomatic at the time of COVID-19 infection. This is not the case for AMI patients. Why ? Biomarkers can potentially vary with the severity of symptoms at the time of infection…

Answer 2: Thank you for your question. Patients in the control post-COVID group could have symptoms of different severity during the SARS-CoV-2 infection but did not have the postcovid symptoms which was assessed by the questionnaire (https://doi.org/10.1002/jmv.27198). Undoubtedly, as it was shown in many studies the level of biomarkers can vary according to the severity of SARS-CoV-2 infection. But the aim of our study was to assess the symptoms of postcovid. Patients that were included in this study suffered mostly from the Omicron strain of SARS-CoV-2 according to epidemiological data in our region during this period. Many patients were not hospitalized as they did not have severe SARS-CoV-2 infection.

Question 3: I could not find any mention of ethical approval and informed consent from the patients throughout the manuscript.

Answer 3: Thank you for your comment. The information about ethical approval and informed consent is in the special sections of the manuscript (Institutional Review Board Statement; Informed Consent Statement) (page 19, line 689; page 20, line 692).

Question 4: Table 1: 12% and 3% of controls received antiplatelet drugs at the time of inclusion which is surprising seems a lot. Can the authors comment about this?

Answer 4: Thank you for your question. Some patients in the control group previously received antiplatelet drugs with no clear indications for them, all antiplatelet drugs were canceled the week before the inclusion in the study. We clarified this information in the Materials and Methods section of the manuscript (page 13, line 397) and Table 1.

Question 5: Patients under anticoagulant drugs were excluded, while those under antiplatelet drugs were included. However, antiplatelet drugs do modify some hemostasis parameters, especially impedance aggregometry. Was this considered in the statistical analysis ?

Answer 5: Thank you for your question! Permanent anticoagulant therapy has a pronounced effect on the activity of plasma hemostasis and makes it almost impossible to interpret the parameters of rotational thromboelastometry (in the NATEM mode) and thrombodynamics. In this regard, the constant use of anticoagulants in our study is an exclusion criterion.

Currently, dual antiplatelet therapy is the cornerstone of the treatment of acute myocardial infarction (https://doi.org/10.1093/eurheartj/ehx393; https://doi.org/10.1093/eurheartj/ehaa575). Almost all patients at the prehospital stage received loading doses of acetylsalicylic acid, and, in the case of ST-segment elevation myocardial infarction, a P2Y12-receptor blocker. Thus, given that the study population is patients with acute myocardial infarction, it is not possible to assess the level of reactivity of their platelets without taking into account antiplatelet therapy, which is one of the limitations of this study.

Prehospital therapy appears to account for the observed differences in arachidonic acid and ADP-induced platelet aggregation between patients with acute myocardial infarction and controls. This fact is included in the Limitations section of the manuscript (page 13; line 372)

Question 6: Were patients/controls free of NSAIDs and SSRIs ?

Answer 6: Thank you for your question. All participants in our study did not receive NSAIDs therapy on a regular basis. We added this information to Table 1. We did not collect data regarding the SSRIs use. We added this information to the Limitations section of the manuscript (page 13, line 371).

Question 7: Methods to measure standard laboratory parameters should be described: reagents and analysers ?

Answer 7: According to your suggestion we added this information to the Supplementary material file.

For blood sampling we used the S-Monovette blood collection system (Sarstedt AG & Co.KG, Germany). We used S-Monovette 2,7 ml K3E (REF 05.1167.001) for complete blood count; S-Monovette 7,5 ml Z-Gel (REF 01.1602.001) for biochemical blood tests; and S-Monovette 5 ml 9NC (REF 05.10I 71.001) for coagulation blood tests.

For ROTEM, thrombodynamics and aggregometry we used S-Monovette 5 ml 9NC (REF 05.10I 71.001). For ROTEM and aggregometry we used whole blood, and for thrombodynamics in both modes we used platelet free plasma obtained by two consecutive centrifugations (1600g x 15 min, RT, then plasma transfer to a new 1,5 ml tube following by 1000g x 5 min, RT).

Plasma used in proteomics study was obtained from whole blood in S-Monovette 2,7 ml K3E (REF 05.1167.001) by centrifugation 4000g x 10 min, RT and then aliquoted and frozen at -80 °C.

A complete blood count (hemoglobin level, red blood cells (RBC), white blood cells (WBC), platelet count, etc.) was performed on Siemens ADVIA  2120i Hematology System with Autoslide (Siemens Healthcare Diagnostics Inc, USA).

Biochemical blood tests (serum creatinine, alanine aminotransferase (ALT), aspartate aminotransferase (AST), etc.) were performed on Siemens ADVIA  2400 Chemistry Analyzer (Siemens Healthcare Diagnostics Inc, USA). Serum creatinine was measured with ADVIA® Chemistry CREA_2 Creatinine Reagents (REF 3039070); ALT was measured with ADVIA® Chemistry ALT(GPT) Reagents (REF 7501976); AST was measured with ADVIA® Chemistry AST(GOT) Reagents (REF 7499718). All reagents for biochemical blood test were from the ADVIA® trade mark (Siemens Healthcare Diagnostics Inc, USA).

Coagulation blood tests (D-dimer, activated partial thromboplastin time (APTT), prothrombin time (PT), international normalized ratio (INR), antithrombin III (AT-III), PT Quick) were performed on ACL TOP 300 CTS (Instrumentation Laboratory, USA). D-dimer was measured with HemosIL D-Dimer HS 500 (REF 00020500100); APTT was measured with HemosIL SynthASil (REF 00020006800); PT was measured with HemosIL RecombiPlasTin 2G (REF 00020003050); AT-III was measured with HemosIL Liquid Antithrombin (REF 00020300400); INR and PT Quick were calculated on ACL TOP 300 CTS (Instrumentation Laboratory, USA). All reagents for coagulation blood tests were from the HemosIL trade mark (Instrumentation Laboratory, USA).

Rotational thromboelastometry was performed in NATEM mode on the ROTEM (The Tem Innovations, GmbH, Germany). For measurement in NATEM mode we used ROTEM® star-tem® 20 (REF 503-10-US) (The Tem Innovations, GmbH, Germany).

Thrombodynamics was conducted on a Thrombodynamics Analyzer System T-2 (HemaCore LLC, Russia) according to the standard technique. For the thrombodynamic test mode we used the Thrombodynamics kit (REF K2-02-10) and for the fibrinolysis test mode we used the Thrombodynamics TDL kit (REF TDL-10). All reagents for thrombodynamic and fibrinolysis tests were from the HemaCore trade mark (HemaCore LLC,Russia and HemaCore SA, Switzerland).

A Multiplate® analyzer (Roche Diagnostics International Ltd, Switzerland) was used to assess impedance aggregometry. For all tests we used Test Cells (REF 06675590); for arachidonic acid (ASPI) platelet activation we used ASPItest reagent (REF 08847533190), for adenosine diphosphate (ADP) platelet activation we used ADPtest reagent (REF 08847550190), and for thrombin receptor-activated peptide-6 (TRAP-6) platelet activation we used TRAPtest reagent (REF 08847509190). All reagents for impedance aggregometry were the Multiplate® analysis trade mark (Roche Diagnostics International Ltd, Switzerland).

This information was added to the Supplementary material file, Materials and Methods (extended) section, page 1-2.

Question 8: I am suprised by the levels or D-dimers that remain completely normal even in the AMI groups. Could the authors comment on this ?

Answer 8: Thank you very much for your question! The issue of using D-dimer as a marker of acute myocardial infarction currently remains a debatable issue. There are a number of studies that have demonstrated an increase in D-dimer level in these patients (https://doi.org/10.1177/147323001003800523; https://doi.org/10.1053/euhj.1999.1519). Some authors also suggested using D-dimer as a marker for risk stratification in patients who have had an acute cardiovascular event (https://doi.org/10.1093/eurheartj/ehab724.1130; https://doi.org/10.1016/j.clinbiochem.2022.02.003). On the other hand, there is a large body of evidence that has not demonstrated a significant increase in D-dimer levels in patients with acute myocardial infarction being a nonspecific marker (https://doi.org/10.1002/ehf2.13027; https://doi.org/10.1186/s12872-019-1239-4; https://doi.org/10.1016/j.ijcard.2020.02.010). In a landmark LIPID trial, the majority of patients had D-dimer levels that were below 500 ng/mL which means that D-dimer levels are low in the majority of patients with acute myocardial infarction (https://doi.org/10.1161/CIRCULATIONAHA.117.029901). In our study, we also did not find any correlations between D-dimer levels and presence of acute myocardial infarction.

Question 9: Table S2: For more clarity, results from TEG, thrombodynamics and aggregometry should be presented separately from proteomics.

Answer 9: Thank you for your suggestion. We divided Table S2 into Table S2 containing hemostatic tests and Table S4 containing proteomics data (Supplementary material, page 14-15, page 18-21).

Question 10: Level of protein S and C4BP are both decreased in AMI-post COVID patients and are positively correlated. The authors thus hypothesize that decreased protein S could be related to the occurence of thrombosis in these patients. This is completely overinterpreted, as protein S measured by proteomics does not reflect its anticoagulant activity and it is well known that C4BP decreases PS activity without changing its antigen level.

Answer 10: According to your suggestion, we excluded this sentence from the Discussion section of the Manuscript.

Question 11: Same remark about coagulation factors measured by proteomics: results should be interpreted with caution as activity might differ from antigen levels…

Answer 11: Thank you for your suggestion. We added this information to the Limitations section and clarified the Conclusions (page 13, line 369; page 15, line 501).  By the activation of coagulation, we mean both the increase in proteins and the results of hemostasis tests.

Question 12: Some parameters presented in Table S1 are not in Table S2. How were parameters chosen ?? Other coagulation factors such as FVIII would have been of great interest..

Answer 12: Thank you for your question! Table S1 include 227 proteins measured by chromato-mass-spectrometry using the panel described previously (https://doi.org/10.1039/C9AN01893J). Further analysis of blood plasma samples from patients revealed 113 proteins that were reliably measured in all samples. The MRM data quality was checked manually in Skyline for all selected proteins/peptides and includes the absence of interference peaks and the good quality of the peak shape, and the ratios of the precursor and product ion. The exemplary MRM data (from Skyline) for selected proteins are presented in Figures S1-S3 in the Supplementary material. All experimental Raw data were uploaded to the PeptideAtlas SRM Experiment Library (PASSEL) and are available via link:

http://www.peptideatlas.org/PASS/PASS04817

Among these proteins, 81 proteins were of interest from their functional role and included in the final analysis (Table S3). These proteins were connected to one of the processes described in the Table S3: hemostasis, extracellular matrix (a structural component or modifier), modification of the state of endothelium, inflammation (except complement system as it was included in a separate group), complement system, lipid metabolism, calcification, and steroid hormone transport (a list of proteins in each group is presented in the Table S3). These processes were chosen because of their direct involvement in cardiovascular disease. Protein functions were estimated manually based on UniProt and NCBI gene databases, and articles in PubMed. We added this information to the Results section (page 5, line 133).

Thank you for your suggestion about coagulation factors. In this work, only coagulation factors IX, XII and XIII (A and B chains) were reliably measured. Coagulation factor VIII was below our limit of detection.

Minor points:

Question 1: Abstract line 27: acute-phase proteins do not grow…

Answer 1: Thank you for your comment! Lipopolysaccharide-binding protein and fibrinogen beta chain are elevated in both AMI groups compared to control groups, and are decreased in AMI post-COVID group compared to AMI control group.

Question 2: Page 1 line 42: the search of biomarkers to stratify cardiovascular risks after COVID-19 infection would have required a propective design (cf supra)

Answer 2: Thank you for your remark! We reformulated this sentence (page 1, line 43).

Question 3: D-dimer results are both in Table 2 and Table S2.

Answer 3: Thank you for your remark. We removed D-dimer from Table S2.

Question 4: Page 15, line 328: antithrombotic drugs include antiplatelet agents. According to Table 1, some healthy volunteers received antiplatelet drugs. Please clarify.

Answer 4: Thank you for your question. We clarified this information in answer to question 4 and updated it in the main text (page 13, line 397) and Table 1.

Question 5: Correlations between biomarkers in the different groups are of no interest and are difficult to interprete (they are not for most of them...) So all the correlation matrices should be moved in the supplemental material.

Answer 5: Thank you for your suggestion! We moved correlation matrices to the Supplementary material file (Figure S4).

Round 2

Reviewer 1 Report

In my opinion, the presented manuscript meets the requirements to be published in IJMS.

Author Response

Thank you

Reviewer 2 Report

ok as is

Author Response

Thank you

Reviewer 3 Report

The authors have addressed the majority of my points except one (the most important).

They should at least add a sentence in the study limitations regarding the retrospective study design and unknown time of infection

Author Response

Question: They should at least add a sentence in the study limitations regarding the retrospective study design and unknown time of infection

Reply: Thank you for your suggestion! We added this information to the Limitation Section of the Manuscript (page 13, line 390).